# ADAPTIVE ESTIMATORS SHOW INFORMATION COMPRESSION IN DEEP NEURAL NETWORKS

**Ivan Chelombiev, Conor J. Houghton & Cian O'Donnell** *
Department of Computer Science
University of Bristol
Bristol, UK
{ic14436,conor.houghton,cian.odonnell}@bristol.ac.uk

## ABSTRACT

To improve how neural networks function it is crucial to understand their learning process. The information bottleneck theory of deep learning proposes that neural networks achieve good generalization by compressing their representations to disregard information that is not relevant to the task. However, empirical evidence for this theory is conflicting, as compression was only observed when networks used saturating activation functions. In contrast, networks with non-saturating activation functions achieved comparable levels of task performance but did not show compression. In this paper we developed more robust mutual information estimation techniques, that adapt to hidden activity of neural networks and produce more sensitive measurements of activations from all functions, especially unbounded functions. Using these adaptive estimation techniques, we explored compression in networks with a range of different activation functions. With two improved methods of estimation, firstly, we show that saturation of the activation function is not required for compression, and the amount of compression varies between different activation functions. We also find that there is a large amount of variation in compression between different network initializations. Secondary, we see that L2 regularization leads to significantly increased compression, while preventing overfitting. Finally, we show that only compression of the last layer is positively correlated with generalization.

## 1 INTRODUCTION

Although deep learning (reviewed by Schmidhuber (2015)) has produced astonishing advances in machine learning (Silver et al., 2017), a rigorous statistical explanation for the outstanding performance of deep neural networks (DNNs) is still to be found.

According to the information bottleneck (IB) theory of deep learning (Tishby & Zaslavsky, 2015; Shwartz-Ziv & Tishby, 2017) the ability of DNNs to generalize can be seen as a type of representation compression. The theory proposes that DNNs use compression to eliminate noisy and task-irrelevant information from the input, while retaining information about the relevant segments (Chechik & Tishby, 2003). The information bottleneck method (Tishby et al., 2000) quantifies the relevance of information by considering an intermediate representation $T$ between the original signal $X$ and the salient data $Y$. $T$ is the most relevant representation of $X$, and is said to be an information bottleneck, when it maximally compresses the input, retaining only the most relevant information, while maximizing the information it shares with the target variable $Y$. Formally, the information bottleneck minimizes the Lagrangian:

$$\mathcal{L}[p(\tilde{t}|x)] = I(T, X) - \beta I(T, Y) \tag{1}$$

where $I(\cdot)$ is mutual information. In this Lagrangian $\beta$ is the Lagrange multiplier, determining the trade-off between compression and retention of information about the target. In the context of deep learning, $T$ is a layer's hidden activity represented as a single variable, X is a data set and Y is the set

*With gratitude to Tilo Burghardt, Rui Ponte Costa, Carl Henrik Ek, Galina Malisheva and Daniil Malkin for their valuable contributions to this research.

of labels. Compression for a given layer is signified by a decrease in $I(T, X)$ value, while $I(T, Y)$ is increasing during training. Fitting behaviour refers to both values increasing.

Shwartz-Ziv & Tishby (2017) visualized the dynamic of training a neural network by plotting the values of $I(T, X)$ and $I(T, Y)$ against each other. This mapping was named the information plane. According to IB theory the learning trajectory should move the layer values to the top left of this plane. In fact what was observed was that a network with *tanh* activation function had two distinct phases: fitting and compression. The paper and the associated talks[1] show that the compression phase leads to layers stabilizing on the IB bound. When this study was replicated by Saxe et al. (2018) with networks using ReLU (Nair & Hinton, 2010) activation function instead of *tanh*, the compression phase did not happen, and the information planes only showed fitting throughout the whole training process. This behaviour required more detailed study, as a constant increase in mutual information between the network and its input implies increasing memorization, an undesired trait that is linked to overfitting and poor generalization (Morcos et al., 2018).

Measuring differential mutual information in DNNs is an ill-defined task, as the training process is deterministic (Saxe et al., 2018). Mutual information of hidden activity $T$ with input $X$ is:

$$I(T, X) = H(T) - H(T|X) \tag{2}$$

If we consider the hidden activity variable $T$ to be deterministic then entropy is:

$$H(T) = -\sum_{i=1}^{N} p_i(t) \log p_i(t) \tag{3}$$

However, if $T$ is continuous then the entropy formula is:

$$H(T) = -\int p_t(t) \log p_t(t) dt \tag{4}$$

In the case of deterministic DNNs, hidden activity $T$ is a continuous variable and $p(T|X)$ is distributed as the delta function. For the delta function:

$$H(T|X) = -\int p_t(t) \log p_t(t) dt = -\infty \tag{5}$$

Thus, the true mutual information value $I(T, X)$ is in fact infinite. However, to observe the dynamics of training in terms of mutual information, finite values are needed. The simplest way to avoid trivial infinite mutual information values, is to add noise to hidden activity.

Two ways of adding noise have been explored previously by Shwartz-Ziv & Tishby (2017) and Saxe et al. (2018). One way is to add noise $Z$ directly to $T$ and get a noisy variable $\hat{T} = T + Z$. Then $H(T|X) = H(Z)$ and mutual information is $I(\hat{T}, X) = H(\hat{T}) + H(Z)$. When the additive noise is Gaussian, the mutual information can be approximated using kernel density estimation (KDE), with an assumption that the noisy variable is distributed as a Gaussian mixture (Kolchinsky & Tracey, 2017). The second way to add noise is to discretize the continuous variables into bins. To estimate mutual information, Shwartz-Ziv & Tishby (2017) and Saxe et al. (2018) primarily relied on binning hidden activity. The noise comes embedded with the discretization that approximates the probability density function of a random variable. In context of neural networks, adding noise can be done by binning hidden activity and approximating $H(T)$ as a discrete variable. In this case $H(T|X) = 0$ since the mapping is deterministic and $I(T, X) = H(T)$.

Generally, when considering mutual information in DNNs, the analyzed values are technically the result of the estimation process and, therefore, are highly sensitive to it. For this reason it is vital to maintain consistency when estimating mutual information. The problem is not as acute when working with DNNs implemented with saturating activation functions, since all hidden activity is bounded. However, with non-saturating functions, and the resulting unbounded hidden activity, the level of noise brought by the estimation procedure has to be proportional and consistent, adapting to the state of every layer of the network at a particular epoch.

In the next section adaptive estimation schemes are presented, both for the binning and KDE estimators. It is shown that for networks with unbounded activation functions in their hidden layers,

---

[1]Available at: `https://youtu.be/FSfN2K3tnJU` and `https://youtu.be/bLqJHjXihK8`

the estimates of information change drastically. Moreover, the adaptive estimators are better able to evaluate different activation functions in a way that allows them to be compared. This approach shows considerable variation in compression for different activation functions. It also shows that L2 regularization leads to more compression and clusters all layers to the same value of mutual information. When compression in hidden layers is quantified with a compression metric and compared with generalization, no significant correlation is observed. However, compression of the last softmax layer is correlated with generalization.

## 1.1 METHODS

In this paper we used a fully connected network consisting of 5 hidden layers with 10-7-5-4-3 units, similar to Shwartz-Ziv & Tishby (2017) and some of the networks described in Saxe et al. (2018). ADAM (Kingma & Ba, 2014) optimizer was used with cross-entropy error function. We trained the network with a binary classification task produced by Shwartz-Ziv & Tishby (2017) for consistency with previous papers. Inputs were 12-bit binary vectors mapped deterministically to the 1-bit binary output, with the categories equally balanced (see Shwartz-Ziv & Tishby (2017) for details). Weight initialization was done using random truncated Gaussian initialization from Glorot & Bengio (2010), with 50 instances of this initialization procedure for every network configuration used. Based on the observed data, even 10 initializations provide a clear picture of the average network behaviour, but for the purpose of consistency with previous studies, we used 50 initializations. 80% of the dataset was used for training, using batches of 512 samples. During the training process, hidden activity for different epochs was saved. The calculation of mutual information was done using the saved hidden activity, after training has been completed; the two processes did not interfere with one another.

## 2 MUTUAL INFORMATION ESTIMATION

The observed dynamics of a neural network's training on the information plane depends strongly on the estimation procedure. Estimating mutual information at different epochs and layers presents a problem as these variables are inherently different from one another. Using more adaptive frameworks makes it possible to choose estimator parameters that would produce comparable estimate values.

## 2.1 ENTROPY-BASED BINNING

Binning networks with saturating activation functions is straightforward, since the saturation points define the range of the binning. With non-saturating activation functions a range of activation values must be specified. In Saxe et al. (2018) for networks with ReLU units in their hidden layer, all hidden activity was binned using a single range, going from zero to $m$, where $m$ is the maximum value any ReLU unit has achieved throughout the training. The argument behind this choice of binning range is that the whole space explored by the ReLU function must be included in the binning process.

This approach is inherently limited, since a network at every epoch is different. If the binning range is determined by picking a maximum value across all epochs and all layers, then the information plane values estimated for a given epoch and a given layer depend on factors that are unrelated to their own properties. In fact, the behaviour of the activation values is markedly different in different layers; the maximum values can differ by as much as two orders of magnitude in different layers and epochs (Figure 1). Using a single maximum value when binning the data usually means that some layers during some epochs will have all of their values binned together in the lowest bin.

To avoid this we use an entropy-based binning strategy (Kohavi & Sahami, 1996) to choose bin boundaries: the bin boundaries are chosen so that each bin contains the same number of unique observed activation levels in a given layer and epoch. When there are repeated activation levels, as can happen for any activation function that is not unbounded, these are ignored when calculating the boundaries. This often occurs near the saturation regions where the activation levels are indistinguishable even as float64 values. However, we calculate the entropy for the whole layer rather than for individual units; in other words, the vector of activation levels of the units in the layer is converted to a vector of bin indices and the entropy is calculated for these vectors. This binning procedure will be referred to as *entropy-based adaptive binning* (EBAB).

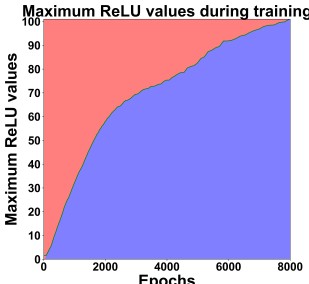

(a) Maximum values in an epoch.

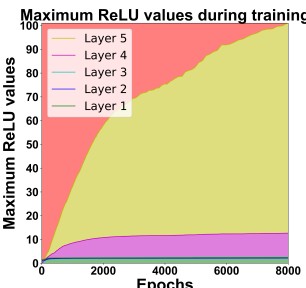

(b) Maximum values for every layer in an epoch.

Figure 1: The graphs show the discrepancy of maximum activation values for a ReLU network. On the left is the maximum value across the whole network, on the right we also show the breakdown by layers. Here the last ReLU layer dominates the maximum values of the whole network. The red area represents the source of inaccuracy brought by binning with non-adaptive range. Similarly, yellow area leads to inaccuracy when all layers are binned using one maximum value.

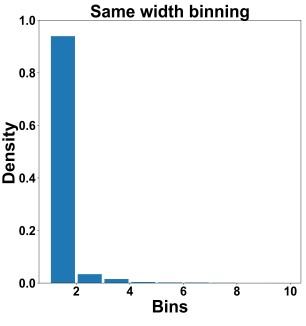

(a) Layer binned with same width bins.

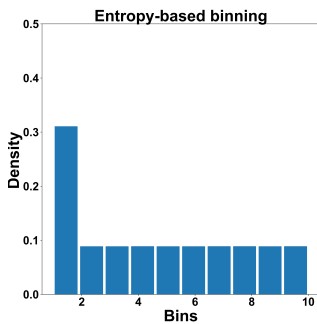

(b) Layer binned with entropy-based binning.

Figure 2: Entropy-based binning allows making accurate mutual information estimates for any distribution of hidden activity, without adjusting the number of bins to the magnitude of the hidden activity.

In this approach there is no need to define a range as part of the binning procedure. It has other advantages. Firstly, the definition of the bins depends on the distribution of the observed activation values, not on the shape of the activation functions. This means this binning strategy applies in the same way to all activation functions, allowing different networks with different activation functions to be compared. Secondly, a fixed width binning approach can be insensitive to the behaviour of the layer because much of the activity occurs in a single bin. An example of binning where a very small subset of hidden activity achieves high values is shown in Figure 2, as such, most of the hidden activity falls into the first bin. With entropy-based binning, bin edges are more densely distributed closer to the lower bound of binning range, providing a better spread.

The difference in mutual information estimates produced with an adaptive binning procedure is significant when visualized on the information plane. Non-adaptive binning consistently underestimates compression. Information planes of the same network in Figure 3 using ReLU analyzed with different estimators show how non-adaptive binning consistently underestimates compression.

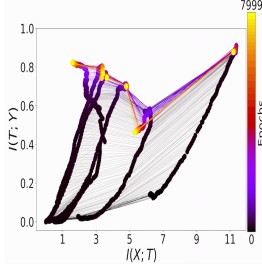 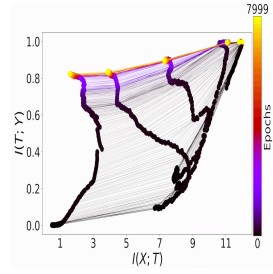

(a) Non-adaptive binning.                    (b) EBAB binning.

Figure 3: One network initialization visualized of the information plane using two different binning estimators. Each line on the information plane represents a single network layer, with colour scheme indicating the epoch during training. Leftmost layer is the output layers, rightmost layer is the closest to the data.

## 2.2 KERNEL DENSITY ESTIMATION

Saxe et al. (2018) used non-parametric KDE estimator outlined by Kolchinsky & Tracey (2017) to estimate mutual information, as well as binning. This method of estimation directly adds small Gaussian noise to the data to produce estimates that are not infinite.

The vast variation of the level of activation values (Figure 1) makes it inappropriate to use a fixed noise level. Due to the difference in activation values, noise added to layers with smaller activation values produces a different effect, as the noise of the same magnitude, added to the largest activation. Hence, KDE estimation with a fixed level of noise produces inconsistent estimations. Here the noise variable $\sigma^2$ was adapted for every layer of every epoch. To implement this we chose a constant reference rate of noise of $\sigma_0^2 = 1 \cdot 10^{-3}$, and then for a given layer $\sigma^2$ was calculated by scaling $\sigma_0^2$ by the maximum value of the hidden activity in that layer for every epoch. With this estimation adjustment, each layer at every epoch received noise of different magnitude, but of similar proportion. This estimation procedure will be referred to as *adaptive* KDE (aKDE).

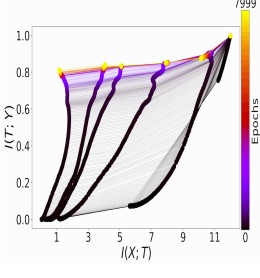 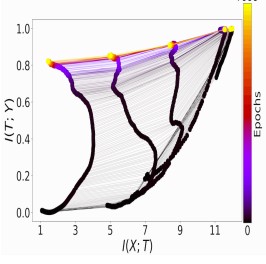

(a) Non-adaptive KDE.                    (b) Adaptive KDE.

Figure 4: Information planes for the same network as in Figure 3, produced with non-adaptive KDE and the adaptive version. (a) Hidden layers show no compression, slight compression visible in the leftmost output layer. (b) Two-penultimate hidden layers undergo compression, as well as the output layer.

Figure 4 demonstrates that scaling the noise variance $\sigma^2$ in unison with the magnitude of the activation values produces estimates that are very similar to EBAB. Similarly, non-adaptive KDE resembles binning with non-adaptive range and widths. The network initialization implemented with ReLU in Figures 3 and 4 exhibits compression when analyzed with EBAB and aKDE. In the next section we look at ReLU networks with different initializations to see whether this compression always happens.

## 3 COMPRESSION IN DEEP NEURAL NETWORKS

After deriving adaptive estimation framework, we investigated the occurrence of compression in networks with non-saturating activation functions. Network initializations showed a large variation of compression behaviour. Figures 3 and 4 showed a network with significant information compression. However, other network initializations can lead to different behaviour on the information plane. Figure 5 shows different ways a network can evolve during training. Based on the observed variation of behaviour, it cannot be said that a network with ReLU always exhibits compression during training, as in the case with the saturating *tanh*. Networks are capable to learn without compression. However, it also cannot be said that using a non-saturating ReLU prevents any compression. Moreover, unlike networks with *tanh* activation function, the compression phase can happen first, with or without subsequent fitting phase.

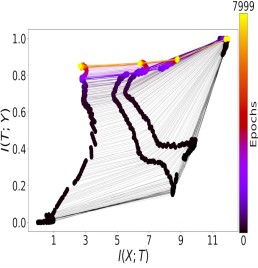
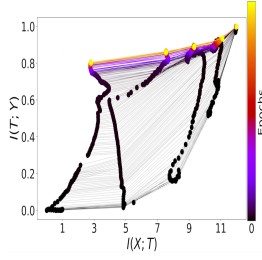
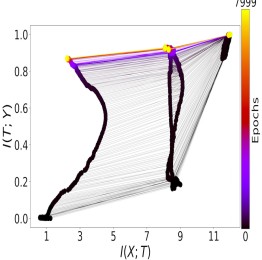

(a) Compression first, then fitting.

(b) No compression, only fitting.

(c) No compression or fitting.

Figure 5: Plots produced with EBAB for the same network configuration using ReLU. The variation of behaviour is caused solely by the stochasticity of the initialization. While the variation of behaviour of the leftmost output layer is modest, the hidden layers have vastly different trajectories on the information planes. ReLU compression happens only in Figure (a) which is marked by an initial leftward movement of two hidden layers. In Figure (b) some fitting is observed in hidden layers, which is signified by a rightward movement. In Figure (c) hidden layers do not compress or fit to data, the $I(X, T)$ values are stationary for hidden layers.

Similarly to previous studies, we generated 50 networks with different random initializations for every activation function to investigate the average behaviour. When one information plot averaged from 50 network initializations is produced, fitting and compression mostly cancel out. On the resulting information plot, the averaged network does not show strong fitting or compression, the $I(T, X)$ values are mostly stable (Figure 6).

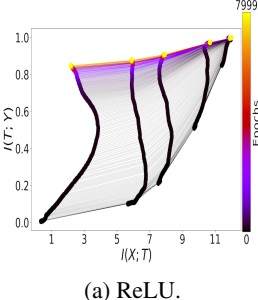
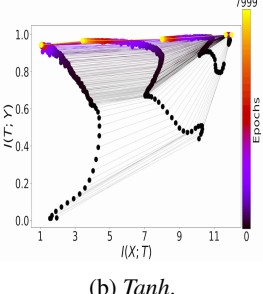

(a) ReLU.

(b) *Tanh*.

Figure 6: The information plane averaged over 50 network initializations using ReLU activation function in the hidden layers. For comparison Figure (b) shows an information plane of an averaged network using saturating *tanh* activation function.

ReLU network averaged over 50 individual network initializations shows no distinct phases of fitting or compression. To see if this behaviour on the information plane is simply the product of absence of a saturating activation function, we have tested other non-saturating activation functions and show information plots averaged over 50 initializations in Figure 7. Different unit types tested are absolute

value, PReLU (He et al., 2015), ELU (Clevert et al., 2015), softplus (Glorot et al., 2011), centered softplus, (Mishkin et al., 2017) and Swish (Ramachandran et al., 2018; Elfwing et al., 2018).

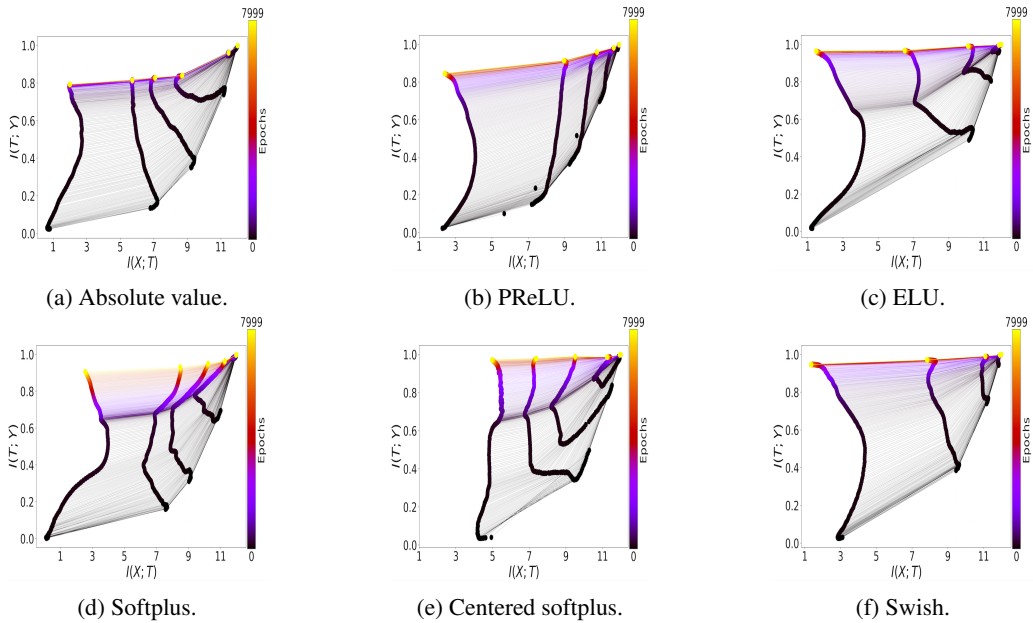

Figure 7: Information planes of networks, using different non-saturating activation functions in the hidden layers. However all networks used softmax function in the output layer. Therefore, the shapes of the leftmost lines on all the information planes show less variation. For every activation function 50 random initializations were trained and averaged mutual information values were used for the information planes presented above.

The information planes demonstrate that finer properties of activation functions have an effect on training dynamics on the information plane, as few functions look similar to each other. Even though *tanh* has the strongest compression, ELU, Swish and centered softplus activation functions also compress information, despite being non-saturating. This compression behaviour is marked by a decrease in $I(X, T)$ values for these activation functions in Figure 7.

## 3.1 L2 REGULARIZATION AND COMPRESSION

We also considered a commonly used technique to increase testing accuracy: L2 regularization. It spreads out the learning by preventing a subset of network activations from having very large values. We implemented L2 regularization on all weights of hidden layers with a network using ReLU.

When we plotted the mutual information estimates on the information plane, L2 regularization induced the network to compress information. Compression occurred in later stages of training, similarly to networks using *tanh*, when networks usually go into an overfitting regime. When looking at information planes averaged over 50 initializations, strong compression is visible. As seen in Figure 8, L2 regularization induces compression in networks with ReLU activation functions, that do not show strong compression on their own. Therefore, compression of information about the input can be a way for a network to avoid overfitting, since networks with L2 regularization increased their test accuracy continuously, without going into an overfitting regime.

Moreover, L2 regularization clusters the mutual information of all layers in a single position on the information planes. Thus it induces more compression in layers that are closer to the data (Figure 8 (d)-(f)). Even with the smallest penalty, compression is distinguishable on the information plane. As the L2 penalty increases, layers are more attracted to a single point on the information plane.

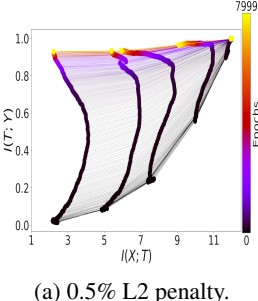 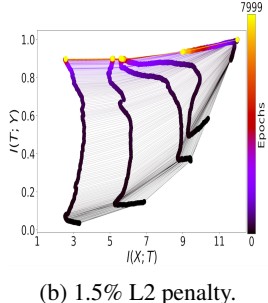 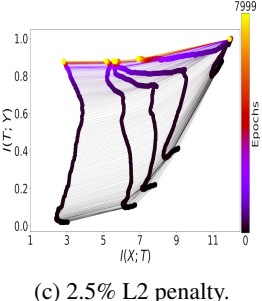

(a) 0.5% L2 penalty.    (b) 1.5% L2 penalty.    (c) 2.5% L2 penalty.

Figure 8: Information plots of a network with ReLU using L2 regularization with different penalties. Information was averaged across multiple initializations. As the penalty increases, the layers get pulled together more strongly.

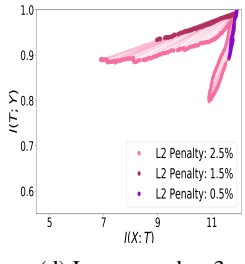 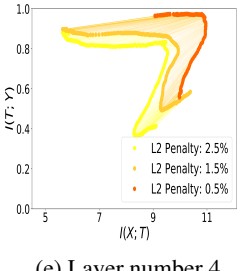 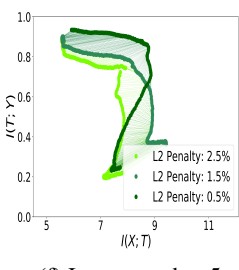

(d) Layer number 3.    (e) Layer number 4.    (f) Layer number 5.

Figure 8: Plots (d)-(g) provide a detailed view of individual layer lines from plots (a)-(c), and are shown on a different scale. For reference, input layer has number 1, output layer is number 7. Colors represent the trajectory transformation that corresponds to an increase in L2 regularization penalty.

## 3.2 COMPRESSION AND GENERALIZATION

In order to analyze compression in a systematic manner, we have developed a simplistic quantitative compression metric to assign a compression score to every network:

$$\text{Score} = \frac{1}{N} \sum_{k=1}^{N} (1 - l_{k,M} / \max_{i \in E}(\mathbf{L}_{k,i})) \tag{6}$$

where $\mathbf{L} \in \mathbb{R}^{N \times M}$ is a matrix containing $I(T, X)$ values of all layers and epochs, $k$ enumerates the layers in $\{1, ..., N\}$, $E$ is the set of epoch indexes $\{1, ..., M\}$. The score ranges from zero to one, with one meaning fully compressed information.

When we compared compression of individual network initializations with accuracy, higher rates of compression did not show a significant correlation with generalization. However, the compression that occurred in the last softmax layer showed significant correlation with accuracy. This is demonstrated in Figure 9a. We also looked at compression of 50 averaged network initializations with different activation functions. The last layer was not taken into account, since it always uses saturating softmax function and is not relevant to comparison of compression in non-saturating activation functions. Similarly to individual ReLU initializations, when compression was compared with the average accuracy of networks, there was no significant correlation, as shown in Figure 9b.

These findings suggest that the information bottleneck is not implemented by the whole network, and generalization can be achieved without compression in the hidden layers. However, the last layer's compression is correlated with generalization and compression there should be encouraged.

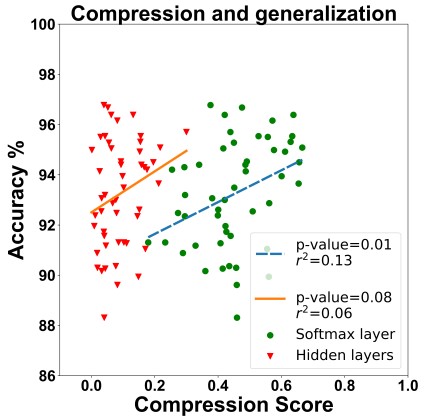
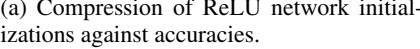
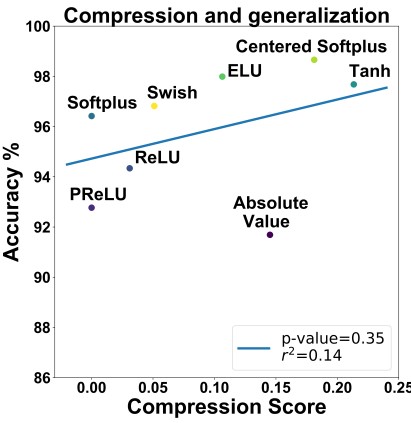

(a) Compression of ReLU network initializations against accuracies.

(b) Compression of various activation functions against accuracy. Values averaged over 50 initializations.

Figure 9: Scatter plots of compression scores compared with accuracies.

# 4 DISCUSSION

In this paper we proposed adaptive approaches to estimating mutual information in the hidden layers of DNNs. These adaptive approaches allowed us to compare behaviour of different activation functions and to observe compression in DNNs with non-saturating activation functions. However, unlike saturating activation functions, compression is not always present and is sensitive to initialization. This may be due to the minimal size of the network architecture that was tested. Experiments with larger convolutional neural networks could be used to explore this possibility.

Different non-saturating activation functions compress information at different rates. While saturation plays a role in compression rates, we show that its absence does not imply absence of compression. Even seemingly similar activation functions, such as softplus and centered softplus, gave different compression scores. Compression does not always happen in later stages of training, but can happen from initialization. Further work is needed to understand the other factors contributing to compression.

We also found that DNNs implemented with L2 regularization strongly compress information, forcing layers to forget information about the input. The clustering of mutual information to a single point on the information plane has never been reported previously. This result could lay the ground for further research to optimize the regularization to stabilize the layers on the information bottleneck bound to achieve better generalization (Achille & Soatto, 2018), as well as linking information compression to memorization in neural networks (Zhang et al., 2016).

There are a few limitations to the analysis presented here. Principally, for tractability, the networks we explored were much smaller and more straightforward than many state of the art networks used for practical applications. Furthermore, our methods for computing information, although adaptive for any distribution of network activity, were not rigorously derived. Finally, our compression metric is ad-hoc. However, overall we have three main observations: first, compression is not restricted to saturating activation functions, second, L2 regularization induces compression, and third, generalization accuracy is positively correlated with the degree of compression only in the last layer and is not significantly affected by compression of hidden layers.

## ACKNOWLEDGMENTS

IC thanks Yobota limited (`https://yobota.xyz`) and Ammar Akhtar for contributing to broadening the reach of this work. CJH thanks the James S. McDonnell Foundation for support through a Scholar Award in Cognition (JSMF #220020239; `https://www.jsmf.org/`). Some of this work was carried out using the computational facilities of the Advanced Computing Research Centre, University of Bristol - `http://www.bris.ac.uk/acrc/`.

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
