# OpenReview forum: "Adaptive Estimators Show Information Compression in Deep Neural Networks"
_ICLR.cc/2019/Conference_

### Official Review · AnonReviewer1 · 2018-10-30

**Rating:** 7
**Confidence:** 3

**Review:**

This paper has 3 principal contributions: it proposes a different way of measuring mutual information in a neural network, proposes a compression score to compare different models, then empirically analyses different activation functions and L2 weights.

This work seems like a welcome addition to the IB thread. To me the most interesting result is simply that activation functions aren't simply about gradient flow, and that they may each have properties that are more or less desirable depending on the domain they might be used on. The authors are careful in the wording of their conclusions, I think with reason; while these results are useful in that there seem to be consistently different behaviors coming from different hyperparameters, information planes show a relatively qualitative part of the picture.

Quantitatively, the proposed compression score is interesting, but as the authors say, simplistic. It seems to me that we care more about the converged models than the whole training trajectory; how does this score evolve with time?

I think an important part of discussion that lacks in this paper is a more in-depth take as to how these findings relate to the Zhang et al [1] memorization vs generalization paper and its follow ups. There seem to be many links to be drawn.

This work is overall a good contribution, but I'll have to agree with the authors' conclusion that more principled analysis methods are required to have a solid grasp of the training dynamics of DNNs. The writing of the paper is good, but the writing of the captions could be improved. (the hard page limit of ICLR is 10 pages and your paper has a lot of captions, so I think investing into a bit more text would be good)


Comments:
- It might be worth to re-explain what the information plane plots are in a figure caption, not just in the text (the text also doesn't really explain that each point is a moment in training, and each thread a different layer, this paper should be readable by someone who has never seen these plots before).
- It's not clear what is going on in figure 5, I can guess but, again, this paper should be readable by anyone in the field. You mention different initializations, but which exactly? What makes you say that 5c has no compression but that 5a does compression first? It should be explained explicitly.
- I believe what you say about Figure 8, but the plots are so similar that it is hard to compare them visually. Maybe a different kind of superposition into a single plot would better illustrate the compression effect of L2?
- Typo in the x axis caption of figures 9.
- Figure 9a is not readable in greyscale (or by a colorblind person), consider using a different symbol for the softmax scatter (and adding this symbol to the legend).
- The first Schmidhuber citation of the paper seems a bit out of place. I think he himself would say that deep learning has been going on for much longer than since 2015. (in fact I think you could just remove the entire first paragraph, it is just unnecessary boilerplate)
- Why should there be a direct correlation between compression and generalization? For example, it is known that training DNNs with soft targets improves test accuracy in classification, or even forcing softness in both targets and representations [2] also improves test accuracy.
- I'm still personally not sold on binning as a strategy to evaluate MI. Did you perform experiments that show that the observed difference is consistent if more computation is done to approximate MI, and not just an artefact of max-entropy binning?

[1] Zhang et al (2016) https://arxiv.org/abs/1611.03530
[2] Verma et al (2018) https://arxiv.org/abs/1806.05236

---

> ### Author Response · Authors · 2018-11-22
> **Review response 1/2**
>
> We are thankful to our referees for their careful and perceptive comments on our paper and are pleased by their positive overall opinion. We have added content to clarify the figures in more detail and addressed minor errors that have been kindly pointed out by the reviewer.
>
> To address your comments individually:
>
> -"Quantitatively, the proposed compression score is interesting, but as the authors say, simplistic. It seems to me that we care more about the converged models than the whole training trajectory; how does this score evolve with time?"
>
> The score compares the most uncompressed state of a layer (largest I(X,T) value observed) and its last observed value. Depending on the amount of fitting this score can stay at zero for a significant portion of training when I(X,T) keeps increasing. If there is little fitting and compression is happening, the score will increase throughout training. Considering the large variety of behaviours that we have observed on the information plane, there is no unique pattern that this score has during training, but it tends to be closer to zero in the initial stages of training.
>
> -"I think an important part of discussion that lacks in this paper is a more in-depth take as to how these findings relate to the Zhang et al [1] memorization vs generalization paper and its follow ups. There seem to be many links to be drawn."
>
> We were careful not to make serious claims about the relationship between information compression and memorization without a thorough investigation. However, we are aware that there could be interesting results from examining this link, and we have mentioned it in the discussion as a potential research opportunity.
>
> -"The writing of the paper is good, but the writing of the captions could be improved. (the hard page limit of ICLR is 10 pages and your paper has a lot of captions, so I think investing into a bit more text would be good)"
>
> We have added more text to the majority of the captions, especially for figures depicting the information plane. In the captions we explicitly comment on the dynamics of networks’ layers, for the ease of interpretability to readers who see these plots for the first time.
>
> - "It might be worth to re-explain what the information plane plots are in a figure caption, not just in the text (the text also doesn't really explain that each point is a moment in training, and each thread a different layer, this paper should be readable by someone who has never seen these plots before). "
>
> We have changed the introduction section to give a better better explanation of how to interpret compression on the information plane, as well as a more explicit explanations in the captions of information plots.
>
> -"It's not clear what is going on in figure 5, I can guess but, again, this paper should be readable by anyone in the field. -You mention different initializations, but which exactly?"
>
> The stochasticity of an initialization has two sources: weight initialization and data shuffling. We have modified the methods section to include information about our initialization procedure: “Weight initialization was done using random truncated Gaussian initialization from Glorot & Bengio (2010), with 50 instances of this initialization procedure for every network configuration used. ... 80% of the dataset was used for training, using batches of 512 samples.”.
>
> -"What makes you say that 5c has no compression but that 5a does compression first? It should be explained explicitly."
>
> Figure 5 shows the information plots of a ReLU network, with different random initialization instances. The spectrum of behaviour presented here aims to demonstrate that fitting-compression phases are not innate to all neural networks, as claimed by previous studies, and there is a significant amount of heterogeneity involved. We have included more explicit explanations about this in the caption to Figure 5.
>
> -"I believe what you say about Figure 8, but the plots are so similar that it is hard to compare them visually. Maybe a different kind of superposition into a single plot would better illustrate the compression effect of L2?"
>
> We agree that it might be time consuming to make out the difference in compression between layers on Figure 8. We have included three extra plots to Figure 8, each one showing only one layer line, with different L2 penalties superposed. The compression is easily distinguished on these plots, while the original plots better demonstrate that these layer lines are pulled to a single point. We have also made sure that these plots are readable in greyscale.
>
> - "Typo in the x axis caption of figures 9".
>
> Thank you, this has been fixed.
>
> - "Figure 9a is not readable in greyscale (or by a colorblind person), consider using a different symbol for the softmax scatter (and adding this symbol to the legend)".
>
> We thank the reviewer for this suggestion. We have reworked this figure to make it more accessible.

---

> ### Author Response · Authors · 2018-11-22
> **Review response 2/2**
>
> - "The first Schmidhuber citation of the paper seems a bit out of place. I think he himself would say that deep learning has been going on for much longer than since 2015. (in fact I think you could just remove the entire first paragraph, it is just unnecessary boilerplate) "
>
> We have put the first paragraph there to provide better context for the paper and to make it slightly more comprehensive for people outside the field of deep learning. We agree that a reference to one paper from 2015 might seem confusing . Although, the paper that we reference is an excellent compilation of research related to the field, so we have changed the citation to say: “ …(reviewed by Schmidhuber (2015))”.
>
> - "Why should there be a direct correlation between compression and generalization? For example, it is known that training DNNs with soft targets improves test accuracy in classification, or even forcing softness in both targets and representations [2] also improves test accuracy."
>
> The parallel between compression and generalization has been drawn originally by Shwartz-Ziv & Tishby (2017), as the paper claimed that the implementation of the information bottleneck is the reason why neural networks are able to perform well. We have not found evidence strong enough to support this claim in general, but we saw a significant correlation between performance and compression of the softmax layer.
>
> - "I'm still personally not sold on binning as a strategy to evaluate MI. Did you perform experiments that show that the observed difference is consistent if more computation is done to approximate MI, and not just an artefact of max-entropy binning?"
>
> We have tried many techniques to estimate MI, including bin-free non-parametric methods like aKDE. The purpose of out paper was to point out potential artefacts with previous binning implementations, then propose a much more flexible binning method (EBAB), which turned out to give similar estimates as aKDE. Also, since hidden activity data was not sparse, binning is as a reliable estimator, when not affected by the magnitude differences of the hidden activity.
> Our main conclusion remains that whichever noise-bearing estimation method is used to measure mutual information in the context of deterministic DNNs, unless the noise is proportional to the magnitude of the hidden activity, MI estimates will be inconsistent, and present a misleading picture on the information plane.

---

### Official Review · AnonReviewer3 · 2018-11-02
**This paper uses information to show compression of DNN with unbounded activation function, but it gives no questions to what form the compression.**

**Rating:** 6
**Confidence:** 4

**Review:**

This paper proposes a method for the estimation of mutual information for networks with unbounded activation functions and the use of L2 regularization to induce more compression.  The use of information planes to study the training behavior of networks is not new.  This paper addresses the issue of unbounded  hidden state activities.  As the differential mutual information in DNN is ill-defined, the authors proposed to add noise to the hidden activity by using the binning process.   It is not clear in the paper that if the binning is applied just for visualizing the information plane or for computing the activities of hidden units in upper layers.   If it is the latter one, it creates unnecessary distortions to the DNN.  As the authors pointed out, different initializations can lead to different behaviour on the information plane.  It would be difficult to draw conclusions based on the experimental results, even they come from the average of 50 individual networks.  Also, the experiences are performed using a particular task, it is not sure if similar behavior is observed in other tasks.   It is, however, more important to understand what makes the compression.   For the L2 regularization, the compression is expected as the regularization tends to limit the values of the  weights.

---

> ### Author Response · Authors · 2018-11-22
> **Review response**
>
> We are thankful to our referees for their careful and perceptive comments on our paper and are pleased by their positive overall opinion.
>
> To address your comments individually:
>
> -"It is not clear in the paper that if the binning is applied just for visualizing the information plane or for computing the activities of hidden units in upper layers."
>
> The calculation of mutual information was done after all training has been done and all hidden activity was saved for every epoch. Binning was used solely for calculating MI after all training has been completed.  We have modified the text (see ‘1.1 Methods’ section) to give a better description of the experimental procedures:
> “During the training process, the hidden activity for different epochs was saved. The calculation of mutual information was done using the saved hidden activity, after training has been completed; the two processes did not interfere with one another.”.
>
> -"It would be difficult to draw conclusions based on the experimental results, even they come from the average of 50 individual networks."
>
> To address our choice of the number of random initializations: when we compared information plane plots using averages over five and ten some variation was visible. However 10 and 25 averaged intializations lead to very similar results. Despite good convergence even at 25 initializations, we used 50 for ease of comparison with previous studies. We have modified the ‘1.1 Methods’ section to include the following: “Based on the observed data, even ten initializations provide a clear picture of the average network behaviour, but for the purpose of consistency with previous studies, we used 50 initializations.”.
>
> -"Also, the experiences are performed using a particular task, it is not sure if similar behavior is observed in other tasks."
>
> In this contribution we set out to study compression in itself, using robust mutual information estimators. We also wanted to see the effect that various activation functions have on this behaviour. That is why we have chosen to use the same dataset as previous study, as discussed in the ‘1.1 Methods’ section.
> We have experimented with other binary tasks and compression was present when MI was measured with adaptive estimators. The results that we have seen were not markedly different from those that we have observed with the main dataset.  Even though the relationship between task structure and strength of compression is a very interesting topic, it was beyond the scope of this investigation.
>
> -"It is, however, more important to understand what makes the compression."
>
> We agree with the reviewer that it could be very interesting to deduce the key precursor to compression and investigate its significance. However, our main effort is focused on outlining mutual information estimates that are robust and applicable to deterministic neural networks regardless of the activation function employed. We were also mainly interested in determining whether compression can happen for non-saturating activation functions in the first place. We believe our paper presents enough evidence to support this effort.

---

### Official Review · AnonReviewer2 · 2018-11-02
**Interesting observations!**

**Rating:** 7
**Confidence:** 4

**Review:**

The authors of this paper studied the popular belief that deep neural networks do information compression for supervised tasks. They studied this compression behavior with tanh and ReLU (and it's variants) activation functions which are saturating and non saturating in nature respectively.

The compression score is computed using Mutual Information Estimation which when computed are usually infinite. For finite mutual information values, noise can be added to hidden activations. For this purpose, two approaches namely Entropy Based Binning(EBAB) and adaptive Kernel Density Estimation(aKDE) were explored. EBAB adds noise to the hidden activations by binning and aKDE by Gaussian noise. Their results show that both EBAB and aKDE exhibit compression in case of ReLU, although this behavior is the strongest in tanh.

Finally, When compression score was plotted against accuracy, higher rates of compression did not show significant correlation with generalization. Hence showing evidence that generalization(or good performance) can be achieved even without information bottleneck(information compression).

Qualms:
1. Figure 7's description that ELU, Swish and centered softplus functions doing compression is not very apparent.
2. Figure 9b: Regression line between compression score and accuracy shows a positive correlation between them. This seems contradictory to the inference.
3. The experiments were done on a 5-layer network with 10-7-5-4-3 nodes respectively on a toy data of 12-bit binary vectors. The study could have included bigger networks with popular datasets which would give substantial support to the trend observed on toy data.

---

> ### Author Response · Authors · 2018-11-22
> **Review response**
>
> We are thankful to our referees for their careful and perceptive comments on our paper and are pleased by their positive overall opinion.
>
> In response to your comments:
>
> -"1. Figure 7's description that ELU, Swish and centered softplus functions doing compression is not very apparent."
>
> According to Shwartz-Ziv and Tishby (2017) information compression is marked by a decrease in I(X,T) values with a simultaneous increase in I(T,Y) during training. We have added a clarification of this in the introduction section: “Compression for a given layer is signified by a decrease in I(T,X) value, while I(T,Y) is increasing.  Fitting behaviour refers to both values increasing.”.
> In Figure 7 we present information plots of networks using different activation functions in their hidden layers. However all used softmax in the output layer, thus the leftward movement of the leftmost layer line should be disregarded. Out of the various activation functions used in the hidden layers, it was ELU, centered softplus and Swish that showed the strongest visible decrease in I(X,T) values, which is also reflected through our compression metric and visualised on Figure 9 (b). We have added extra clarification to the caption of Figure 7: “Information planes of networks, using different non-saturating activation functions in the hidden layers. However all networks used softmax function in the output layer. Therefore, the shapes of the leftmost lines on all the information planes show less variation. For every activation function 50 random initializations were trained and averaged mutual information values were used for the information planes presented above.”.
>
> -"2. Figure 9b: Regression line between compression score and accuracy shows a positive correlation between them. This seems contradictory to the inference."
>
> The p-value for the regression line on Figure 9 (b) was 0.35, too large to claim that there is significant correlation. However, we agree that «generalization accuracy is … not affected by compression of hidden layers» in the discussion section could be interpreted as misleading, given the provided evidence. We apologise for this inexactness and we have changed the wording to «not significantly affected by the compression of the hidden layers».
>
> -"3. The experiments were done on a 5-layer network with 10-7-5-4-3 nodes respectively on a toy data of 12-bit binary vectors. The study could have included bigger networks with popular datasets which would give substantial support to the trend observed on toy data."
>
> The main constraint to using larger networks (CNN with MNIST, for example) is the computational costs of mutual information calculations, which exceed the computation required for the network training, as well as memory costs of storing hidden activity for hundreds of epochs. Computational requirements for mutual information estimation for even a modest CNN are a few orders of magnitude greater than those for the configuration that we have been experimenting with. Given that we have observed strong sensitivity of information plane behaviour to the randomness of initialization, the resources required to reproduce these findings (using 50 initializations for every activation function) with larger networks were intractable for this project, while presenting an information plot for a single initialization could be misleading due to high variability of behaviour.
> Presented with the trade-off between studying CNNs using only one random initialization or smaller networks with a multitude of intializations, which can show average behaviour, we have made the decision to do the latter. As outlined in the methods section, the chosen configuration has also provided the ease of comparison with previous studies that used it.
> However, we are hopeful that hardware limitations will be less prevalent in further studies.

---

> > ### Comment · AnonReviewer2 · 2018-11-30
> > **Reply to Rebuttal**
> >
> > Thank you for the rebuttal. The modifications proposed do address my concerns. Also, I do agree that the scale of experiments should not be the only factor for evaluating the quality of an article. I am moving my score to 7 hoping to see more grounded results and extensions of the presented work.

---

### Author Response · Authors · 2018-11-22
**Revision**

We are grateful to the reviewers for their detailed feedback on our submission.

We have uploaded a revised script. It includes three new graphs in Figure 8, a reworked Figure 9, more details about the experimental setup in the "1.1 Methods" section. Captions to figures that demonstrate information planes have been expanded to include more explanations.

---

### Meta-Review · Area_Chair1 · 2018-12-13
**Interesting visualizations, but more rigor and analysis would help**

**Confidence:** 3
**Recommendation:** Accept (Poster)

**Metareview:**

This paper suggests that noise-regularized estimators of mutual information in deep neural networks should be adaptive, in the sense that the variance of the regularization noise should be proportional to the range of the hidden activity. Two adaptive estimators are proposed: (1) an entropy-based adaptive binning (EBAB) estimator that chooses the bin boundaries such that each bin contains the same number of unique observed activation levels, and (2) an adaptive kernel density estimator (aKDE) that adds isotropic Gaussian noise, where the variance of the noise is proportional to the maximum activity value in a given layer. These estimators are then used to show that (1) ReLU networks can compress, but that compression may or may not occur depending on the specific weight initialization; (2) different nonsaturating noninearities exhibit different information plane behaviors over the course of training; and (3) L2 regularization in ReLU networks encourages compression. The paper also finds that only compression in the last (softmax) layer correlates with generalization performance. The reviewers liked the range of experiments and found the observations in the paper interesting, but had reservations about the lack of rigor in the paper (no theoretical analysis of the convergence of the proposed estimator), were worried that post-hoc addition of noise distorts the function of the network, and felt that there wasn't much insight provided on the cause of compression in deep neural networks. The AC shares these concerns, and considers them to be more significant than the reviewers do, but doesn't wish to override the reviewers' recommendation that the paper be accepted.